# The effectiveness of language nursing intervention on mental health in children with poor language skills

Xi Shu[1], Yingzi Xiao[2], Lingzhi Yang [3]*

1 Department of Cardiovascular Medicine, Anhua County People's Hospital, Anhua County, Yiyang City, Hunan Province, China, 2 Radiomedical Imaging Center, Anhua County People's Hospital, Anhua County, Yiyang City, Hunan Province, China, 3 Department of Alcohol Addiction and Internet Addiction, Hunan Brain Hospital (Hunan Second People's Hospital), Changsha City, Hunan Province, China

* 15084976952@163.com, yang15084976952@gmail.com

## Abstract

### Background

Mental health issues in adulthood often start in childhood, so it's important to identify these issues early and find ways to manage them. To our knowledge, no study was found that evaluated the long-term effects of language nursing intervention on mental health in children with poor language skills. This study, therefore, aimed to evaluate the long-term impact of a language nursing intervention on the mental health of children with poor language skills.

### Methods

We estimated poor language skills prevalence in 3-4-year-old children who were planning to enter kindergartens in Hunan, China. After selecting these children, we divided them into two experimental and control groups. The experimental group received a nursing intervention related to language skills for eight months. After eight months, the language skills of both groups were re-evaluated. Then, in the follow-up evaluation, the mental health of these children was evaluated at the ages of 9–10 years. Univariate and multivariate regression models adjusted with sampling weights were used to estimate the correlation of mental health and risk factors.

### Results

The language skills of the experimental group increased significantly compared to before the protocol (from 87.4±10.87 to 98.08±7.13; p = 0.001). At the end of the eight-month nursing intervention, the language skills of the experimental group were significantly higher than the control group (98.08±7.13 in experimental group and 87.51±9.59 in control group; p = 0.001). In multivariate analysis, single-parent family and not participating in the nursing protocol related to language skills at the age of 3–4 years were related to high symptoms of depression, anxiety and stress symptoms (single-parent family: for depression symptoms, OR = 1.28, 95% CI = 0.88–1.42; for stress symptoms, OR = 1.31, 95% CI = 0.79–2.74 and for anxiety symptoms, OR = 1.42, 95% CI = 0.97–2.44; not participating in the nursing

---

**Data Availability Statement:** The "minimal data set" of study is present entirely within manuscript.

**Funding:** The author(s) received no specific funding for this work.

**Competing interests:** The authors have declared that no competing interests exist.

protocol related to language skills at the age of 3–4 years: for depression symptoms, OR = 2.98, 95% CI = 1.80–5.19; for stress symptoms, OR = 1.88, 95% CI = 1.23–2.01 and for anxiety symptoms, OR = 2.67, 95% CI = 1.51–3.77; p<0.05).

## Conclusion

The current study showed the effectiveness of this intervention on both language skills and mental health of children with poor language skills.

## Introduction

Childhood and adolescence mental health issues often signal lifelong recurring problems that continue into adulthood. These disorders cause major psychological distress and financial burdens for families and society. In recent years, mental disorders among childhood and adolescence are on the rise. According to the 2019 Global Burden of Disease study, the prevalence of mental disorders in children and young people was 6.8 to 13.95% [1]. Also, about 20% of kids and teenagers aged 3–17 in the United States have some kind of mental, emotional, developmental, or behavioral issue [2].

This makes them a big health deal in China [3]. Li et al (2019) found that 17.5% of Chinese children and adolescents had mental disorders [4]. There are various reasons for the psychological problems of childhood and adolescence, including their speech, language and communication problems. Mental health and social-emotional well-being challenges that occur at a later stage in life can caused by the existence of speech and language impairments during childhood. These impairments put children and adults at risk of developing mental health problems [5]. Also, people who have communication disorder as their primary disability are highly likely to suffer from psychological complications, mainly anxiety as well as depression [5]. Good communication skills make relationships thrive and boost mental health. Talking about feelings, solve issues, handling conflicts, and making choices through speech, language, and communication skills help people live better lives. However, when people struggle to communicate, it can mess up their whole world—from friendships to school to work to living on their own. It even affects how they fit in with their community [6].

A great deal of research has been conducted on this subject. A longitudinal study done by the University of Bristol revealed that pragmatic language needs in childhood were connected with psychotic experiences in adolescence [7]. In an Australian study published in 2018, they found that preschool children with persistent language needs and unstable language development patterns were more likely to develop social-emotional and behavioral challenges [8]. A literature review in 2002 established that emotional or behavioral diagnoses accompanied over half (57%) children diagnosed with a broad spectrum of language issues [9]. Multiple studies also indicate that children facing difficulties related to languages and literacy have more trouble with attention and socializing than other kids [9, 10]. In a recent study in China, the prevalence of language disorders in children was 8.5% [11]. This number was equal to 7.6% in England [12].

Considering that many mental problems in adulthood have their roots in childhood, therefore, it is necessary to diagnose these problems in children and provide an intervention solution to control them. In most studies, the short-term effects of these interventions have been evaluated. For example, Qiang (2023) reported that after 6 months of speech therapy with language cognitive and emotional speech community, the ability to speak and communicate in

children with speech disorders improved [13]. To our knowledge, no study was found that evaluated the long-term effects of language nursing intervention on mental health in children with poor language skills. Therefore, the present study aimed to evaluate the long-term impact of a language nursing intervention on the mental health of children with poor language skills.

## Materials and methods

### Study design and participants

This study uses a multi-stage, stratified, cluster random sampling design to select sample of kindergarten children in Hunan, China. We divided districts into smaller sections called primary sampling units and further split these into secondary sampling units based on whether the kindergartens were public or private and their quality level. In China, kindergartens are categorized as either public or private. Also, kindergartens are rated from highest to lowest based on factors like teaching quality, health care, facilities, management, and more. The highest rating is "demonstration" and the lowest is "third level." To match kindergartens, we considered only the first and second level public kindergartens.

From each secondary sampling unit, we randomly chose one or two kindergartens, and all children in these kindergartens were invited to participate in our evaluation. This led to a group of 6,820 children aged 3–4 years from selected kindergartens. We did not include kindergartens for children with special needs, so children with severe autism, moderate to severe intellectual disabilities, and permanent hearing loss were not part of our study. This study consists of two parts. In the first part, which was a pretest-posttest semi-experimental method with control group, an 8-month nursing intervention was used (February 1 to September 30, 2018 with 657 participants). In the next step and with a 6-year follow-up, re-evaluations were carried out (September 2024 with 642 participants) (Fig 1).

This study was performed in line with the principles of the Declaration of Helsinki. Approval was granted by Clinical Ethics Review of Hunan Brain Hospital (Hunan Second Peoples Hospital) (2017Ethics-31). All parents were told about the study's goals and procedures, and they agreed to let their children participate by signing a consent form.

### Procedures

In first evaluation (2018), the children' language skills were checked using the Diagnostic Receptive and Expressive Assessment of Mandarin (DREAM) [14]. This questionnaire has already been used in China [11]. The reliability and validity of this questionnaire in the present study was good (Cronbach's alpha: 0.78). After the test, they got a total score for language and four other scores (receptive, expressive, semantic, and syntax). If a child scored below 80 on any of these five parts of the DREAM test, they were seen as having poor language skills.

Children who had other health issues or special needs, according to what their parents said, were not included in the study. To see how well the Children could think without using language, they were administered the Raven Standard Progressive Matrices Questionnaire [15]. This questionnaire has already been used in China [11]. The reliability and validity of questionnaire in this study was good (Cronbach's alpha: 0.709). Children who have both limited language skills and lower intelligence levels (with an IQ below 70) were not included.

After children with low language skills were selected, their parents were informed that they could refer their child to Rehabilitation and Training Center that the researchers considered. The children who visited this center received a nursing programs including language cognitive rehabilitation training and emotional intervention for eight months (one session per week, each session lasting 30 minutes). This nursing protocol has been used in previous research [13]. Children who did not participate in the protocols for more than 3 sessions were excluded

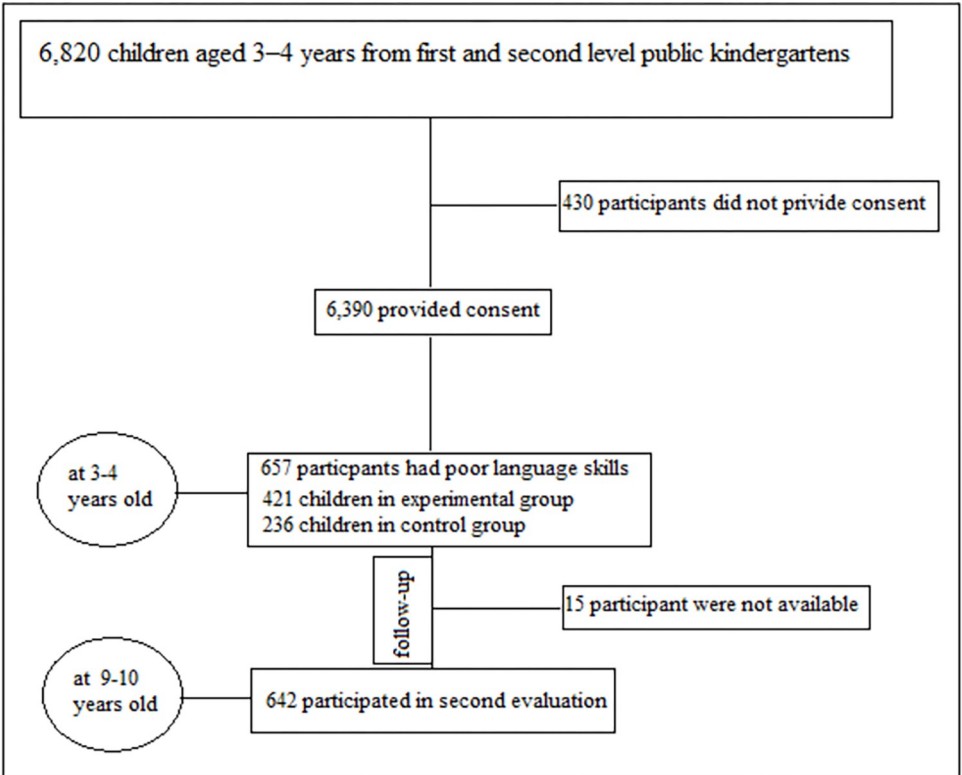

**Fig 1. Flow chart of included analytical sample.**

from the experimental research group. Children with low language skills who did not participate in the nursing protocol were also considered as the control group.

After eight months of the nursing protocol, the language skills of children in both groups (the group that received the nursing protocol and the group that did not participate in this protocol) were measured by DREAM again.

Children were also evaluated at age 9–10 (2024). If during this time interval (from age 3–4 to age 9–10), each of the experimental and control group participants participated in any nursing protocol related to language, they were excluded from the research.

At this stage, the sociodemographic information and mental health status over the preceding week of children were evaluated.

The first part of the survey asked for information such as age, gender, type of family they live in, and their parents' education level. A single-parent family was defined as living only with one parent.

The second part of the survey looked at how the participant's mental health status in the past week. It used tests like the Center for Epidemiology Studies Depression scale (CES-D) and the Depression Anxiety Stress Scale (DASS-21) to measure mental health status.

The CES-D is a 4-point Likert scale that measures the prevalence of depressive symptoms. Higher scores mean more severe depressive symptoms. The CES-D scale in Chinese has been tested and found to be reliable in past studies with Chinese groups [16, 17]. A score of 20 or more on the CES-D was used to identify if someone has depressive symptoms.

The DASS-21 as a 4-point Likert scale was used to measure three negative feelings such as stress, anxiety, and depression. Since the symptoms of depression were already assessed using the CES-D, the study only used two parts or 14 questions of the DASS-21 to check the

participants' stress and anxiety levels. The total score for each part was found by doubling the sum of its seven questions. Higher scores showed severe negative feelings. Anxiety symptoms were identified with scores over 7, and stress symptoms were identified with scores over 14 [17, 18]. The reliability of both the CES-D and DASS-21 was good, with Cronbach's alpha 0.801 and 0.919, respectively, showing strong internal consistency.

### Statistical analysis

Categorical variables are shown as percentages, and continuous variables are given as Mean ± SD. The independent and dependent tests were used to compare intra- and inter-group differences. We first used univariate logistic regression to find variables that were statistically important. Then, we used a multivariate logistic regression, to see how variables were correlated to depressive, anxiety, and stress symptoms. In this study, we used IBM SPSS Statistics 26 and considered results to be statistically significant if the P-value was less than 0.05.

## Results

Out of a total of 6820 eligible participants, 6390 children aged 3–4 participated in the study. Using the DREAM Questionnaire, it was found that 10.28% of children (n = 657) have low language skills (50.59% female). 421 children fully participated in the 8-month nursing protocol (52.02% female). Therefore, they were considered as the experimental group. 236 children were considered as control group (49.15% female). In pre-intervention (age 3–4), 17.96% of the participants had a single-parent family, which reached 19% in the follow-up (age 9–10). The 39.12% paternal and 40.33% maternal education was >12 years before the intervention. In the follow-up, this number increased to 42.99% and 43.46% in fathers and mothers, respectively (Table 1).

After 8 months of intervention, the language skills of the experimental group participants increased significantly compared to before the protocol (from 87.4±10.87 to 98.08±7.13; p = 0.001). At the end of the eight-month nursing protocol period, the language skills of the experimental group were significantly higher than the control group (98.08±7.13 in experimental group and 87.51±9.59 in control group; p = 0.001) (Table 1).

At 9–10 years old, the participants who were present in the previous phase of the study (at 3–4 years old) took part in the new assessment. 15 of participants (5 from the experimental group and 10 from the control group) did not participate in this evaluation for various reasons. In total, 642 participants were evaluated in this phase. It was seen that 13.1% exhibited depression symptoms, 22.5% demonstrated stress signs, and 6.9% exhibited anxiety symptoms (Table 1).

In univariate analysis, single-parent family was found to be related to higher depression, stress, and anxiety symptoms (for depression symptoms, OR = 1.14, 95% CI = 1.10–2.19; for stress symptoms, OR = 1.09, 95% CI = 1.02–1.21 and for anxiety symptoms, OR = 1.12, 95% CI = 0.99–1.16; p<0.05) (1). Also, not participating in the nursing protocol related to language skills at the age of 3–4 years was found to be related to higher depression, stress, and anxiety symptoms (for depression symptoms, OR = 2.21, 95% CI = 1.32–5.11; for stress symptoms, OR = 1.92, 95% CI = 1.02–2.34 and for anxiety symptoms, OR = 2.05, 95% CI = 1.33–3.19; p<0.05) (Table 1).

In multivariate analysis, these two variables (single-parent family and not participating in the nursing protocol related to language skills at the age of 3–4 years) were related to high symptoms of depression, anxiety and stress symptoms (single-parent family: for depression symptoms, OR = 1.29, 95% CI = 0.88–1.42; for stress symptoms, OR = 1.31, 95% CI = 0.79–2.74 and for anxiety symptoms, OR = 1.42, 95% CI = 0.97–2.44; not participating in the

**Table 1. Characteristics of participants in pre-intervention, post-intervention and follow-up.**

| | | Pre-intervention | | | Post-intervention | | | Follow-up | | |
|---|---|---|---|---|---|---|---|---|---|---|
| | | All | Experimental | Control | All | Experimental | Control | All | Experimental | Control |
| | | (n = 657) | (n = 421) | (n = 236) | (n = 657) | (n = 421) | (n = 236) | (n = 642) | (n = 416) | (n = 226) |
| Gender (N, %) | Male | 322, 49.41% | 202, 47.98% | 120, 50.85% | | | | 315, 49.06% | 200, 48.08% | 115, 50.88% |
| | Female | 335, 50.59% | 219, 52.02% | 116, 49.15% | | | | 327, 50.94% | 216, 51.92% | 111, 49.12% |
| Single-parent family (N, %) | Yes | 118, 17.96% | 68, 16.15% | 50, 21.18% | | | | 122, 19% | 71, 17.06% | 51, 22.56% |
| | No | 539, 82.04% | 353, 83.85% | 186, 78.82% | | | | 520, 81% | 345, 82.94% | 175, 77.44% |
| Paternal education (N, %) | ≤12 years | 400, 60.88% | 257, 61.04% | 143, 60.59% | | | | 366, 57.01% | 246, 59.14% | 120, 53.09% |
| | >12 years | 257, 39.12% | 164, 38.96% | 93, 39.41% | | | | 276, 42.99% | 170, 40.86% | 106, 46.91% |
| Maternal education (N, %) | ≤12 years | 392, 59.67% | 254, 60.33% | 138, 58.47% | | | | 366, 56.54% | 236, 56.73% | 130, 57.52% |
| | >12 years | 265, 40.33% | 167, 39.67% | 98, 41.53% | | | | 279, 43.46% | 180, 43.27% | 99, 42.48% |
| language skills (Mean ± SD) | | | 87.4±10.87 | 86.5±9.2 | | 98.08±7.13*# | 87.51 ±9.59 | | | |
| Depression (N, %) | Yes | | | | | | | 84, 13.1% | 37, 8.89% | 47, 20.8% |
| | No | | | | | | | 558, 86.9% | 379, 91.11% | 179, 79.2% |
| Stress (N, %) | Yes | | | | | | | 144, 22.5% | 83, 19.95% | 61, 26.99% |
| | No | | | | | | | 498, 77.5% | 333, 80.05% | 165, 73.01% |
| Anxiety (N, %) | Yes | | | | | | | 44, 6.9% | 21, 5.04% | 23, 10.17% |
| | No | | | | | | | 598, 93.1% | 395, 94.96% | 203, 89.83% |

\* = Significant difference with pre-intervention

\# = Significant difference with the control group

nursing protocol related to language skills at the age of 3–4 years: for depression symptoms, OR = 2.98, 95% CI = 1.80–5.19; for stress symptoms, OR = 1.88, 95% CI = 1.23–2.01 and for anxiety symptoms, OR = 2.67, 95% CI = 1.51–3.77; p<0.05). The Nagelkerke R square value in depression, stress, and anxiety models was 0.212, 0.249, and 0.205, respectively. This mean that single-parent family and not participating in the nursing protocol related to language skills at the age of 3–4 years would explain 21.2%, 24.9%, and 20.5% of the variation in depression, stress, and anxiety in the final adjusted model (Table 2).

## Discussion

Developmental language disorder is a common condition that affects how children develop language skills. It has significant and long-lasting impacts on personal growth. Some studies in the UK and China showed that about 7.6% and 8.5% of children aged 3 to 5 have developmental language disorder [11, 12]. This is more common than autism spectrum disorder (which affects 1.86% of children) and attention-deficit/hyperactivity disorder (ADHD, which affects 5% of children). The prevalence of poor language skills in the present study was 10.28%. Also, this disorder poses a greater risk for children's school and daily life. For example, children with ADHD are reported to be two to three times more likely to have serious spelling issues,

**Table 2. Univariate and multivariate regression analysis of mental health status risk factors.**

| | | Depression symptoms | | Stress symptoms | | Anxiety symptoms | |
|---|---|---|---|---|---|---|---|
| | | OR, 95% CI | p | OR, 95% CI | p | OR, 95% CI | p |
| Gender | | | | | | | |
| Male | Univariate | 1.00 (Ref) | | 1.00 (Ref) | | 1.00 (Ref) | |
| | Multivariate | | | | | | |
| Female | Univariate | 1.01, (0.95–1.02) | 0.321 | 0.94, (0.83–0.91) | 0.119 | 0.99, (0.93–0.98) | 0.581 |
| | Multivariate | 0.90, (0.89–1.00) | 0.219 | 0.98, (0.90–0.97) | 0.202 | 1.02, (0.89–0.95) | 0.399 |
| Single-parent family | | | | | | | |
| No | Univariate | 1.00 (Ref) | | 1.00 (Ref) | | 1.00 (Ref) | |
| | Multivariate | | | | | | |
| Yes | Univariate | 1.14, (1.10–2.19) | 0.01 | 1.09, (1.02–1.21) | 0.003 | 1.12, (0.99–1.16) | 0.01 |
| | Multivariate | 1.28 (0.88–1.42) | <0.001 | 1.31(0.79–2.74) | <0.001 | 1.42 (0.97–2.44) | <0.001 |
| Paternal education | | | | | | | |
| ≤12 years | Univariate | 1.00 (Ref) | | 1.00 (Ref) | | 1.00 (Ref) | |
| | Multivariate | | | | | | |
| >12 years | Univariate | 0.93, (0.91–0.99) | 0.091 | 0.97, (0.93–1.02) | 0.616 | 0.95, (0.82–0.88) | 0.232 |
| | Multivariate | 0.98, (1.01–1.11) | 0.104 | 0.88, (0.98–1.12) | 0.291 | 1.12, (1.07–1.15) | 0.202 |
| Maternal education | | | | | | | |
| ≤12 years | Univariate | 1.00 (Ref) | | 1.00 (Ref) | | 1.00 (Ref) | |
| | Multivariate | | | | | | |
| >12 years | Univariate | 0.89, (0.79–0.85) | 0.599 | 0.93, (0.90–0.97) | 0.395 | 0.88, (0.89–0.96) | 0.202 |
| | Multivariate | 0.75, (0.92–1.02) | 0.11 | 0.87, (0.92–0.98) | 0.449 | 0.95, (1.04–1.13) | 0.128 |
| Participation in nursing intervention at 3-4years old | | | | | | | |
| Yes | Univariate | 1.00 (Ref) | | 1.00 (Ref) | | 1.00 (Ref) | |
| | Multivariate | | | | | | |
| No | Univariate | 2.21, (1.32–5.11) | <0.001 | 1.92 (1.02, 2.34) | <0.001 | 2.05 (1.33, 3.19) | <0.001 |
| | Multivariate | 2.98, (1.80–5.19) | <0.001 | 1.88 (1.23, 2.01) | <0.001 | 2.67 (1.51, 3.77) | <0.001 |

dyslexia, and trouble learning math [19]. In contrast, children with developmental language disorder are four to six times more likely to face these issues [20]. Therefore, quick diagnosis and use of appropriate interventions can prevent some children's problems school and daily life. One of the results of current study was that a nursing intervention related to language skills for eight months could improve the language skills of children with low language skills. These results are consistent with other studies [13, 21, 22]. Considering that children with autism spectrum disorder and attention-deficit/hyperactivity disorder were not included in the present study, it seems that the use of nursing interventions related to language skills can improve the language skills of children with poor language skills.

Some studies showed that having language problems during childhood is associated with a high probability of mental problems in the later periods of life [23–25]. Considering the effectiveness of the nursing intervention used in this study on improving language skills, another aim of the current study was to evaluate the long-term effects of this intervention on mental health. This follow-up study is one of the first studies that evaluated the long-term effects of a nursing intervention related to language skills on children's mental health. The result of the present study indicated that not using nursing intervention related to language skills in 3-4-year-old children with poor language skills was associated with high symptoms of depression, anxiety and stress at 9–10 years old.

In a study, it was found that about one-ten of young individuals with developmental language disorder had very few or no symptoms of internalizing (e.g., anxiety and depression)

[26]. Additionally, these individuals often experience a mix of symptoms from different types of problems, such as issues like feeling emotional or having trouble with friends happen together as they grow from childhood to adolescence [26].

It seems that peer problems, bullying victimization, and maladaptive emotional regulation strategies are among the reasons for high mental problems in young people with developmental language disorder [27, 28].

In addition to the relationship of not using nursing intervention related to language skills, single-parent was also related to children's mental health. The present study showed that the symptoms of depression, stress and anxiety were higher in participants with single-parent households. This is consistent with other studies [29, 30]. A single parent is someone who is raising a child or children on their own. This could be a single mother or a single father. They might be a single parent because they got divorced, their partner passed away, or they separated from their partner. Sometimes, a single parent is the result of an unplanned pregnancy. Other times, a person might choose to become a single parent, either by using donor sperm or adopting a child. In these cases, the person lives with their child or children without the help of another adult to share the responsibilities of parenting [31, 32]. Globally, about 6.8% of children live with just one parent. In China, the number of single parents with children under 18 has increased from 3.9% in 2001 to 5.9% in 2016. Among these single parents in China, 78% are mothers raising their children by themselves [33]. Understanding why children raised by single parents are more likely to have problems adjusting is important for research. However, studies on this topic have been few. It seems that this could be because parents and children sometimes struggle to communicate well and might not adjust to their situations as well as they could [34, 35].

This study, like many studies, has limitations. In this study we considered only the first and second level public kindergartens. Therefore, caution should be used in interpreting the results to all children. Another limitation of the current study is that only children with poor language skills were evaluated long-term and we cannot compare the mental health of these children with children with normal language skills. It is suggested that in another study, a long-term comparison be made in the mental health of people who had language disorders in childhood (and were treated) with healthy children. Another limitation of the present study is that it was conducted in a city in China. Considering the cultural and economic differences between different cities in China, one should be cautious in generalizing the results of this study to children in other cities. Therefore, it is suggested to conduct a wider study in China.

The purpose of this study was to evaluate the short-term and long-term effectiveness of a nursing intervention related to language skills on the language skills and mental health of children with poor language skills. The current study showed the effectiveness of this intervention on both language skills and mental health of these children. Therefore, it is suggested to evaluate language skills at a young age and at the stages of children's entry into kindergarten, and if there is a problem, appropriate interventions should be used so that these children can avoid possible problems during their life. Also, considering the existence of a relationship between single-parent and mental health problems in children, it is suggested that the government has a supporting role for these families so that children can benefit from mental health. It is also suggested to the medical centers and kindergartens that, in cooperation with each other, children with language problems are diagnosed as quickly as possible first, and in the next stage, appropriate interventions are provided by the medical centers and especially by the nurses.

## Author Contributions

**Conceptualization:** Xi Shu, Yingzi Xiao, Lingzhi Yang.

**Data curation:** Xi Shu, Yingzi Xiao, Lingzhi Yang.

**Formal analysis:** Yingzi Xiao, Lingzhi Yang.

**Investigation:** Lingzhi Yang.

**Methodology:** Xi Shu, Lingzhi Yang.

**Project administration:** Lingzhi Yang.

**Resources:** Xi Shu.

**Supervision:** Yingzi Xiao, Lingzhi Yang.

**Validation:** Lingzhi Yang.

**Visualization:** Lingzhi Yang.

**Writing – original draft:** Xi Shu, Yingzi Xiao, Lingzhi Yang.

**Writing – review & editing:** Lingzhi Yang.

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
