## [Decision Letter · Decision Letter 0]

16 Sep 2024

PONE-D-24-32947The effectiveness of language nursing intervention on mental health in children with poor language skillsPLOS ONE

Dear Dr. Yang,

Thank you for submitting your manuscript to PLOS ONE. After careful consideration, we feel that it has merit but does not fully meet PLOS ONE’s publication criteria as it currently stands. Therefore, we invite you to submit a revised version of the manuscript that addresses the points raised during the review process.

We look forward to receiving your revised manuscript.

Kind regards,

Jennifer Coto, PhD

Academic Editor

PLOS ONE

**Journal Requirements:**

2. We note that your Data Availability Statement is currently as follows: All relevant data are within the manuscript and its Supporting Information files

**Additional Editor Comments:**

I agree with the limitations noted by the reviewers, particularly the revisions to methodology and results. Please address these concerns.

Reviewers' comments:

Reviewer's Responses to Questions

**Comments to the Author**

1. Is the manuscript technically sound, and do the data support the conclusions?

Reviewer #1: Yes

Reviewer #2: Yes

Reviewer #3: Yes

2. Has the statistical analysis been performed appropriately and rigorously? 

Reviewer #1: Yes

Reviewer #2: Yes

Reviewer #3: Yes

3. Have the authors made all data underlying the findings in their manuscript fully available?

Reviewer #1: No

Reviewer #2: Yes

Reviewer #3: No

4. Is the manuscript presented in an intelligible fashion and written in standard English?

Reviewer #1: Yes

Reviewer #2: Yes

Reviewer #3: Yes

5. Review Comments to the Author

**Reviewer #1: **The study entitled "The Effectiveness of Language Nursing Intervention on Mental Health in Children with Poor Language Skills" aims to evaluate the long-term impact of a language nursing intervention on the mental health of children exhibiting poor language skills. The findings presented in this paper offer valuable insights into the effectiveness of nursing interventions that address language skills and their correlation with the mental health of affected children. However, several critical aspects warrant further attention.

Firstly, the study employs various measures, and it is imperative to address the reliability and validity of these instruments. Additionally, while the participant selection procedures are well-articulated, the demographic backgrounds of the participants are not adequately detailed. It is essential to include a table summarizing critical background information for the participants across both groups, as well as the results of their respective measures. Such information should encompass the total number of children who participated in the study, the number of children from single-parent households, the educational backgrounds of the parents, and descriptive statistics such as means, medians, or ranges for the various measures employed.

Given that this is a longitudinal study and that the results suggest that family structure, particularly single-parent status, plays a significant role in the mental health outcomes for children, it is necessary to inquire whether there have been any changes in the marital status of the participants' families over the course of the study. Furthermore, clarification is needed regarding how such changes were addressed in the data analysis.

**Reviewer #2: **The article presents a valuable contribution to the literature on the relationship between language skills and mental health in children. The study design, data analysis, and discussion of findings are generally well-executed. However, there are a few areas where the article could be strengthened.

There are a few minor grammatical or stylistic issues that could be addressed, but they do not significantly impact the overall readability or clarity of the text.

Introduction

Line 35: The statistic of 17.5% of Chinese children and adolescents with mental disorders could be further contextualized by providing information about the prevalence of mental health disorders in other countries or regions.

Lines 62-65: The mention of Qiang (2023) could be strengthened by providing more details about the specific findings of that study regarding the short-term effects of language nursing intervention.

Methods

Line 70: The phrase "semi-experimental method" could be further clarified to specify the type of semi-experimental design used (e.g., pretest-posttest with control group).

Lines 73-78: The description of the multi-stage, stratified, cluster random sampling design is clear and well-explained. However, it might be helpful to add a visual diagram or flowchart to illustrate the process.

While the sample size of 6,820 children is substantial, it might be helpful to provide additional information about the distribution of participants across different demographic variables (gender, socioeconomic status).

If there were any dropouts during the study, it would be important to report the dropout rate and reasons for attrition.

Procedures

There are a few minor inconsistencies or typos: For example, in line 93, "questionnaire" should be capitalized as "Questionnaire."

Line 98: Consider rephrasing "they took the Raven Standard Progressive Matrices test" to "they were administered the Raven Standard Progressive Matrices test."

Lines 113-114: The phrase "from 3-4 to 9-10 years old" could be clarified to "from age 3-4 to age 9-10."

Line 124: "four-point Likert scale" could be clarified as "a 4-point Likert scale."

Statistical analysis

In addition to statistical significance, it would be beneficial to report effect sizes (e.g., odds ratios) to quantify the magnitude of the relationships between variables.

It would be helpful to provide information about the model fit (e.g., R-squared, Nagelkerke R-squared) to assess the overall performance of the regression models.

It is important to check the assumptions of the logistic regression models (e.g., linearity, multicollinearity) to ensure the validity of the results.

Results

While the statistical significance of the findings is clear, reporting effect sizes (e.g., odds ratios) would provide a more complete picture of the magnitude of the relationships between variables.

It would be helpful to discuss the clinical significance of the findings, particularly in terms of the practical implications for intervention and prevention.

Discussion

Paragraphs 1-3: The discussion of the prevalence of developmental language disorder and its association with mental health challenges is clear and well-supported by literature.

Paragraphs 4-5: The discussion of the study's findings regarding the effectiveness of the nursing intervention and the relationship between single-parent families and mental health is well-organized and easy to follow.

Paragraphs 2-3: The comparison of developmental language disorder with autism spectrum disorder and ADHD is accurate and provides valuable context.

Paragraph 4: The discussion of the limitations of the study is balanced and acknowledges potential areas for future research.

The discussion could be further strengthened by explicitly outlining the practical implications of the findings for clinicians working with children with language disorders.

The limitations could be followed by specific recommendations for future research to address these limitations and further explore the relationship between language skills and mental health.

**Reviewer #3:** Please refer to the attached file for the complete review of the research article titled "The Effectiveness of Language Nursing Interventions on Mental Health in Children with Poor Language Skills.

6. PLOS authors have the option to publish the peer review history of their article (what does this mean?). If published, this will include your full peer review and any attached files.

Reviewer #1: **Yes: **Pei-Hua Chen

Reviewer #2: No

Reviewer #3: **Yes: **Bashar M Farran

---

## [Author Response · Author response to Decision Letter 0]

27 Sep 2024

Dear editor

- The “minimal data set” of study is present entirely within manuscript

- The list of authors and affiliations are correct.

Reviewer 1: 

the study employs various measures, and it is imperative to address the reliability and validity of these instruments.

Answer: The reliability and validity of the questionnaires were entered in the Procedures section.

demographic backgrounds of the participants are not adequately detailed”

answer: I added table 1. 

Given that this is a longitudinal study and that the results suggest that family structure, particularly single-parent status, plays a significant role in the mental health outcomes for children, it is necessary to inquire whether there have been any changes in the marital status of the participants' families over the course of the study. Furthermore, clarification is needed regarding how such changes were addressed in the data analysis.

answer: I added table 1.

Reviewer 2:

The statistic of 17.5% of Chinese children and adolescents with mental disorders could be further contextualized by providing information about the prevalence of mental health disorders in other countries or regions.

Answer: I added to introduction (lines 34-37). 

The mention of Qiang (2023) could be strengthened by providing more details about the specific findings of that study regarding the short-term effects of language nursing intervention.

Answer: The type of intervention, its duration and the participants are specified.: lines 64-65. 

Line 70: The phrase "semi-experimental method" could be further clarified to specify the type of semi-experimental design used (e.g., pretest-posttest with control group).

Answer: I added. Lines: 84-88. 

Lines 73-78: The description of the multi-stage, stratified, cluster random sampling design is clear and well-explained. However, it might be helpful to add a visual diagram or flowchart to illustrate the process. 

Answer: I added in study design section. 

While the sample size of 6,820 children is substantial, it might be helpful to provide additional information about the distribution of participants across different demographic variables (gender, socioeconomic status).

Answer: I added table 1. 

If there were any dropouts during the study, it would be important to report the dropout rate and reasons for attrition.

Answer: In the first stage, some participants did not have written consent. In the follow-up phase, we could not find 15 participants. It is clear in Figure 1.

Procedures

There are a few minor inconsistencies or typos: For example, in line 93, "questionnaire" should be capitalized as "Questionnaire."

Line 98: Consider rephrasing "they took the Raven Standard Progressive Matrices test" to "they were administered the Raven Standard Progressive Matrices test."

Lines 113-114: The phrase "from 3-4 to 9-10 years old" could be clarified to "from age 3-4 to age 9-10."

Line 124: "four-point Likert scale" could be clarified as "a 4-point Likert scale."

Answer: I edit it. Thanks.

In addition to statistical significance, it would be beneficial to report effect sizes (e.g., odds ratios) to quantify the magnitude of the relationships between variables.

It would be helpful to provide information about the model fit (e.g., R-squared, Nagelkerke R-squared) to assess the overall performance of the regression models.

It is important to check the assumptions of the logistic regression models (e.g., linearity, multicollinearity) to ensure the validity of the results.

Answer: I added to results section. 

Results

While the statistical significance of the findings is clear, reporting effect sizes (e.g., odds ratios) would provide a more complete picture of the magnitude of the relationships between variables.

It would be helpful to discuss the clinical significance of the findings, particularly in terms of the practical implications for intervention and prevention.

Answer: I added to results section. 

Reviewer3: The literature review could be more comprehensive. While it cites

relevant studies, it does not adequately explain the gap this research fills compared to

previous studies. The authors mention that little research has been done on long-term effects

but do not detail what has been done, leaving the gap somewhat vague.

Answer: I corrected the sentence. Lines 66-67. I also added new content in the introduction section. Lines: 34-37. 

Sample Size and Generalizability:

Answer: I added the limitation that the study was conducted in only one city in lines 247-252 and also provided a suggestion for future study.

Questions/Areas for Further Research and future direction:

Answer: Given the limitations, I have provided suggestions for future articles. Lines: 247-253, 261-264.

---

## [Decision Letter · Decision Letter 1]

18 Oct 2024

The effectiveness of language nursing intervention on mental health in children with poor language skills

PONE-D-24-32947R1

Dear Dr. Yang,

We’re pleased to inform you that your manuscript has been judged scientifically suitable for publication and will be formally accepted for publication once it meets all outstanding technical requirements.

Kind regards,

Jennifer Coto, PhD

Academic Editor

PLOS ONE

Additional Editor Comments (optional):

Reviewers' comments:

Reviewer's Responses to Questions

**Comments to the Author**

1. If the authors have adequately addressed your comments raised in a previous round of review and you feel that this manuscript is now acceptable for publication, you may indicate that here to bypass the “Comments to the Author” section, enter your conflict of interest statement in the “Confidential to Editor” section, and submit your "Accept" recommendation.

Reviewer #2: All comments have been addressed

Reviewer #3: All comments have been addressed

2. Is the manuscript technically sound, and do the data support the conclusions?

Reviewer #2: Yes

Reviewer #3: Yes

3. Has the statistical analysis been performed appropriately and rigorously? 

Reviewer #2: Yes

Reviewer #3: Yes

4. Have the authors made all data underlying the findings in their manuscript fully available?

Reviewer #2: Yes

Reviewer #3: Yes

5. Is the manuscript presented in an intelligible fashion and written in standard English?

Reviewer #2: Yes

Reviewer #3: Yes

6. Review Comments to the Author

Reviewer #2: Thank you for submitting the revised version of your manuscript and for your replies to my previous comments and suggestions. Upon reviewing the revised manuscript, I am pleased to confirm that you have adequately addressed the points I raised.

Reviewer #3: (No Response)

7. PLOS authors have the option to publish the peer review history of their article (what does this mean?). If published, this will include your full peer review and any attached files.

Reviewer #2: No

Reviewer #3: **Yes: **Bashar M Farran

---

## [Editor Report · Acceptance letter]

23 Oct 2024

PONE-D-24-32947R1 

PLOS ONE

Dear Dr. Yang, 

I'm pleased to inform you that your manuscript has been deemed suitable for publication in PLOS ONE. Congratulations! Your manuscript is now being handed over to our production team.

Kind regards, 

on behalf of

Dr. Jennifer Coto 

Academic Editor

PLOS ONE